# Original antigenic sin responses to *Betacoronavirus* spike proteins are observed in a mouse model, but are not apparent in children following SARS-CoV-2 infection

**Stacey A. Lapp**[1,2], **Venkata Viswanadh Edara**[1,2,3], **Austin Lu**[1,2], **Lilin Lai**[1,2,3], **Laila Hussaini**[1,2], **Ann Chahroudi**[1,2], **Larry J. Anderson**[1,2], **Mehul S. Suthar**[1,2,3], **Evan J. Anderson**[1,2,4], **Christina A. Rostad**[1,2]*

1 Department of Pediatrics, Emory University School of Medicine, Atlanta, GA, United States of America, 2 Center for Childhood Infections and Vaccines, Children's Healthcare of Atlanta and Emory University School of Medicine, Atlanta, GA, United States of America, 3 Yerkes Primate Center, Emory University, Atlanta, GA, United States of America, 4 Department of Medicine, Emory University School of Medicine, Atlanta, GA, United States of America

* Christina.rostad@emory.edu

**Data Availability Statement:** All relevant data are within the manuscript and its Supporting Information files.

## Abstract

### Background

The effects of pre-existing endemic human coronavirus (HCoV) immunity on SARS-CoV-2 serologic and clinical responses are incompletely understood.

### Objectives

We sought to determine the effects of prior exposure to HCoV *Betacoronavirus* HKU1 spike protein on serologic responses to SARS-CoV-2 spike protein after intramuscular administration in mice. We also sought to understand the baseline seroprevalence of HKU1 spike antibodies in healthy children and to measure their correlation with SARS-CoV-2 binding and neutralizing antibodies in children hospitalized with acute coronavirus disease 2019 (COVID-19) or multisystem inflammatory syndrome (MIS-C).

### Methods

Groups of 5 mice were injected intramuscularly with two doses of alum-adjuvanted HKU1 spike followed by SARS-CoV-2 spike; or the reciprocal regimen of SARS-Cov-2 spike followed by HKU1 spike. Sera collected 21 days following each injection was analyzed for IgG antibodies to HKU1 spike, SARS-CoV-2 spike, and SARS-CoV-2 neutralization. Sera from children hospitalized with acute COVID-19, MIS-C or healthy controls (n = 14 per group) were analyzed for these same antibodies.

### Results

Mice primed with SARS-CoV-2 spike and boosted with HKU1 spike developed high titers of SARS-CoV-2 binding and neutralizing antibodies; however, mice primed with HKU1 spike

**Funding:** This work was funded by a Center for Childhood Infections and Vaccines (CCIV) pilot award from Children's Healthcare of Atlanta and Emory University School of Medicine (to C.A.R.) and a Fast Grant from Emergent Ventures at the Mercatus Center at George Mason University (to A. C.). The funders had no role in study design, data collection and analysis, decision to publish, or preparation of the manuscript.

**Competing interests:** E.J.A. has received personal fees from AbbVie, Pfizer, and Sanofi Pasteur for consulting, and his institution receives funds to conduct clinical research unrelated to this manuscript from MedImmune, Regeneron, PaxVax, Pfizer, GSK, Merck, Novavax, Sanofi-Pasteur, Janssen, and Micron. He also serves on a safety monitoring board for Sanofi- Pasteur and Kentucky BioProcessing, Inc. C.A.R.'s institution has received funds to conduct clinical research unrelated to this manuscript from BioFire Inc, GSK, MedImmune, Micron, Janssen, Merck, Moderna, Novavax, PaxVax, Pfizer, Regeneron, Sanofi-Pasteur. She is co-inventor of patented RSV vaccine technology unrelated to this manuscript, which has been licensed to Meissa Vaccines, Inc. These do not alter our adherence to PLOS ONE policies on sharing data and materials.

and boosted with SARS-CoV-2 spike were unable to mount neutralizing antibodies to SARS-CoV-2. HKU1 spike antibodies were detected in all children with acute COVID-19, MIS-C, and healthy controls. Although children with MIS-C had significantly higher HKU1 spike titers than healthy children (GMT 37239 vs. 7551, $P$ = 0.012), these titers correlated *positively* with both SARS-CoV-2 binding (r = 0.7577, $P$<0.001) and neutralizing (r = 0.6201, $P$ = 0.001) antibodies.

## Conclusions

Prior murine exposure to HKU1 spike protein completely impeded the development of neutralizing antibodies to SARS-CoV-2, consistent with original antigenic sin. In contrast, the presence of HKU1 spike IgG antibodies in children with acute COVID-19 or MIS-C was not associated with diminished neutralizing antibody responses to SARS-CoV-2.

## Introduction

As the coronavirus disease 2019 (COVID-19) pandemic continues and the first vaccinations are administered, our understanding of the immune responses to SARS-CoV-2 continues to evolve. A prevailing question has been what role pre-existing immunity to endemic human coronaviruses (HCoVs) plays in the serologic responses to SARS-CoV-2. The *Betacoronaviruses*, HKU1 and OC43, and *Alphacoronaviruses*, 229E and NL63, cause seasonal respiratory illnesses in both adults and children worldwide. Seroprevalence data indicate that infection with HCoVs occurs during early childhood [1], and the majority of adults are seropositive with antibody titers that wane over time [2, 3]. Cross-reactive antibodies are elicited within genera, but less so between *Alphacoronaviruses* and *Betacoronaviruses* [4]. Cross-reactive antibodies are predominantly directed against non-neutralizing antigens, including the S2 subunit of the spike protein and nucleocapsid proteins. Because children have more frequent exposures to HCoVs, some have hypothesized that pre-existing cross-reactive immunity to HCoVs may in part explain the reduced COVID-19 disease severity observed in children [5–7].

Although SARS-CoV-2 is a *Betacoronavirus*, it shares only 33% amino acid identity with the HCoV *Betacoronaviruses* within the spike protein, which is the predominant immunogen and neutralizing antigen of coronaviruses. Thus, although some pre-pandemic sera do have SARS-CoV-2 spike-reactive antibodies, these antibodies are poorly neutralizing [8]. The widespread seroprevalence of SARS-CoV-2 binding, non-neutralizing antibodies has led to concern that pre-existing immunity from prior HCoV exposures may contribute to aberrant serologic responses to the antigenically similar SARS-CoV-2, as in original antigenic sin (OAS). OAS refers to the preferential induction of antibodies directed against an original, priming antigen rather than a structurally similar boosting antigen. The mechanism of OAS is thought to be attributable to the initial development and differentiation of memory B cells directed against the original antigen of exposure. Upon secondary exposure to a structurally similar antigen, these memory B cells undergo clonal expansion to preferentially produce antibodies directed against the original priming antigen. The phenomenon of OAS has been well described with several other viruses including dengue and influenza, and has led to concerns about its impact on disease severity and vaccine development [9].

In this study, our objectives included determining if prior exposure to HKU1 spike protein in a mouse model impacted serologic responses to SARS-CoV-2 upon spike protein challenge.

We further sought to determine the baseline seroprevalence of HKU1 spike antibodies in healthy children and to assess if the presence or titer of HKU1 antibody affected the response to SARS-CoV-2 binding and neutralizing antibodies in children hospitalized with acute COVID-19 or multisystem inflammatory syndrome (MIS-C).

## Experimental results

### Murine results

Two groups of five BALB/c mice were injected intramuscularly (IM) with 10 μg each of alum-adjuvanted full-length spike protein antigens according to the schedule shown in Fig 1. One group received a prime (d0) and boost (d21) with SARS-CoV-2 full-length spike protein, followed by a prime (d42) and boost (d63) with HCoV HKU1 full-length spike protein. The second group received a reciprocal immunization regimen of a prime and boost with HKU1 spike followed by SARS-CoV-2 spike prime and boost. Submandibular bleed samples were collected at 3 weeks following each injection, and a terminal bleed was performed 3 weeks following the last injection.

We measured IgG antibody titers to SARS-CoV-2 spike and HKU1 spike antigens in the two groups of mice by ELISA. We also measured the neutralizing antibody titers of these plasma samples using a Focus Reduction Neutralization Test (FRNT) with an infectious clone of SARS-CoV-2 virus (Wuhan strain, A.1 based on PANGO lineage) as previously described

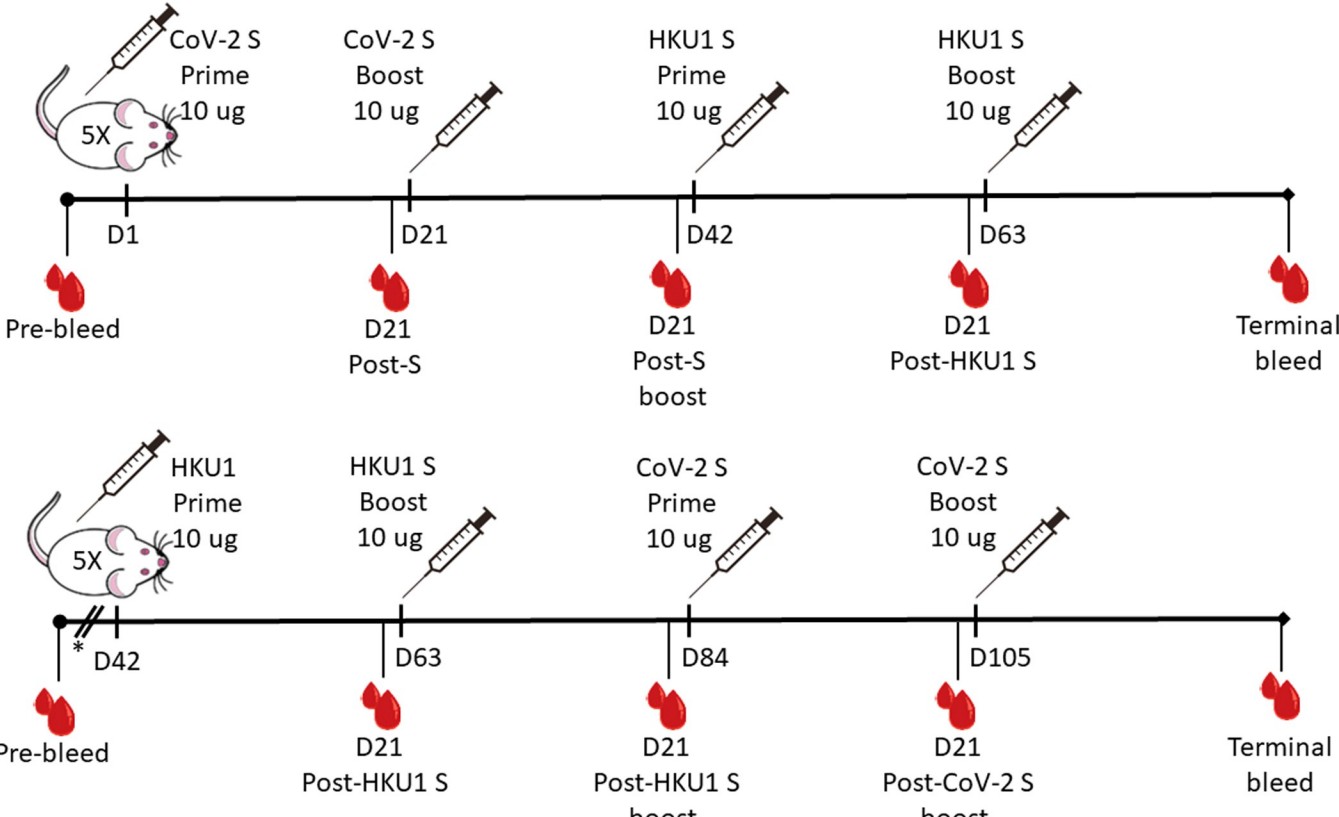

**Fig 1. Schematic of intramuscular spike protein administrations in groups of five BALB/c mice.** Group 1 received prime and boost with SARS-CoV-2 spike, followed by prime and boost with HKU1 spike. Group 2 received a reciprocal administration regimen, with prime and boost with HKU1 spike, followed by prime and boost by HKU1 spike. D, days post-administration; S, spike; SARS-CoV-2, severe acute respiratory syndrome coronavirus 2. *These mice were immunized with nucleocapsid protein 21 and 42 days prior to utilization for this study.

[10, 11] (Fig 2 and S1 Dataset). As expected, the SARS-CoV-2 spike-primed mice developed SARS-CoV-2 spike IgG antibodies that significantly increased in titer after the first injection (1914 vs. 85, $P = 0.012$) and peaked at 21 days after the SARS-CoV-2 spike boost (139316 vs 85, $P < 0.001$, Fig 2A). The SARS-CoV-2 spike IgG antibody titers did not further increase after subsequent administration of HKU1 spike. The HKU1 spike-primed mice also generated cross-reactive anti-SARS-CoV-2 spike IgG antibodies that reached statistical significance on day 21 post-HKU1 spike boost (2594 vs. 85, $P = 0.018$, Fig 2B). Subsequent injection with SARS-CoV-2 spike incrementally boosted SARS-CoV-2 spike IgG titers, which increased significantly after the second boost (51523 vs. 2594, $P = 0.031$). These findings suggested that

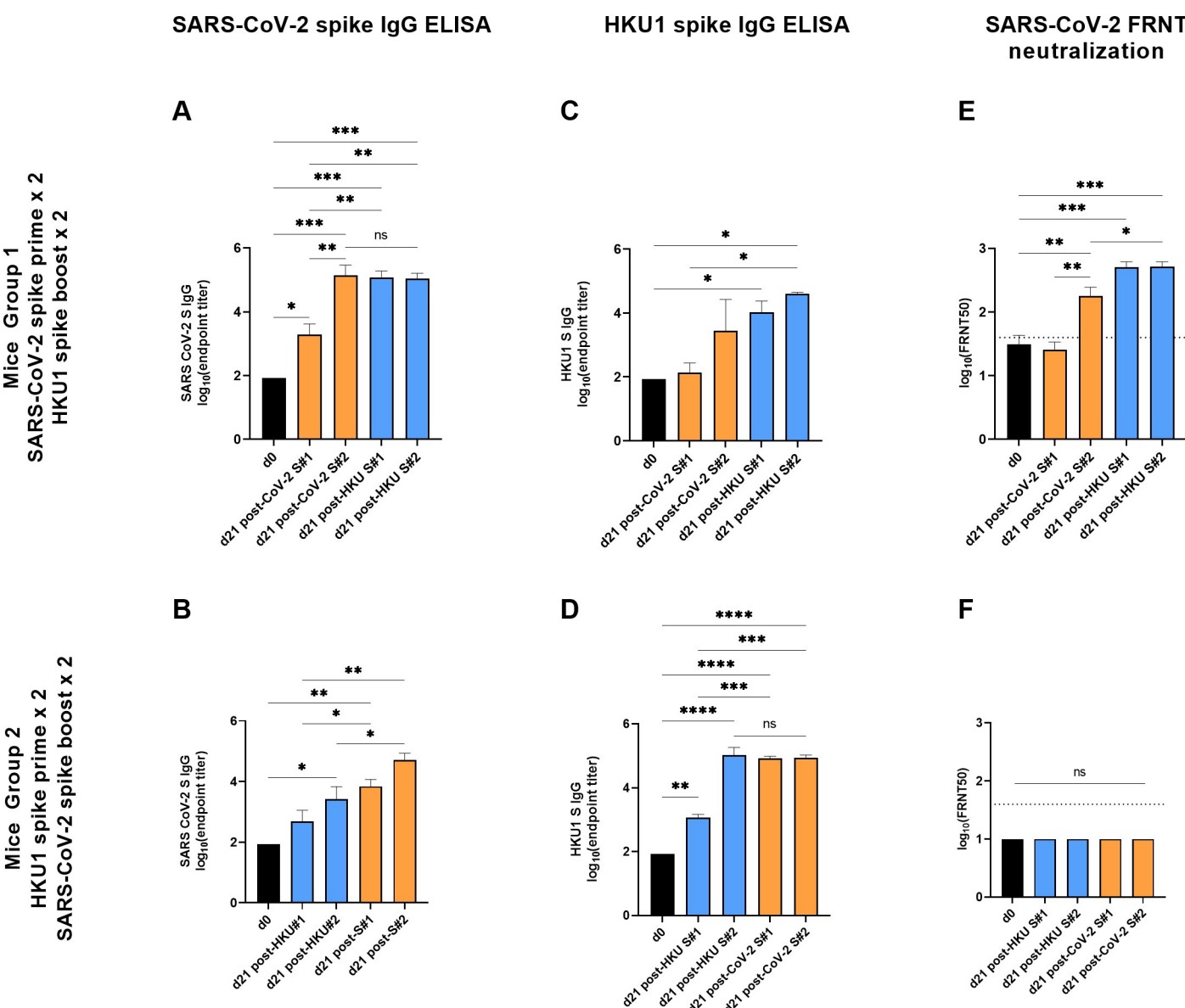

**Fig 2. Priming mice with HKU1 spike protein prior to boosting with SARS-CoV-2 spike protein completely impeded the development of SARS-CoV-2 neutralizing antibodies.** SARS-CoV-2 (A,B) and HKU1 (C,D) full-length spike IgG binding and SARS-CoV-2 neutralizing (E, F) antibodies in mice are shown as log (end-point titer). Group 1 was primed with two doses of alum-adjuvanted SARS-CoV-2 spike and boosted with two doses of alum-adjuvanted HKU1 spike (A, C, E). Group 2 received the reciprocal regimen of HKU1 spike prime and SARS-CoV-2 spike boost (B, D, F). $^*P < 0.05$; $^{**}P < 0.01$; $^{***}P < 0.005$; $^{****}P < 0.001$.

heterologous boosting drove affinity maturation against related epitopes on the boosting SARS-CoV-2 spike antigen.

Analogously, SARS-CoV-2 spike-primed mice developed cross-reactive HKU1 spike binding IgG antibodies, but these did not significantly increase above baseline following either the first (137 vs. 85, $P = 0.990$) or second injection (2851 vs. 85, $P = 0.116$, Fig 2C). Subsequent immunization with HKU1 spike protein increased the HKU1 spike IgG titers significantly over baseline (10351 vs. 85, $P = 0.038$) which were maintained after HKU1 spike boost (39719 vs. 10351, $P = 0.750$, Fig 2C). As expected, the HKU1 spike-primed mice generated HKU1 spike-binding IgG titers that increased significantly after the initial HKU1 injection (1197 vs. 85, $P = 0.001$) and peaked following HKU1 boost (106167 vs. 85, $P<0.001$, Fig 2D). Subsequent injection with SARS-CoV-2 spike protein did not further increase HKU1 spike IgG titers. The plateau in antibody titers observed following repeated boosting is a known phenomenon whereby antibody feedback limits further B cell expansion by masking immunodominant epitopes [12].

We then performed SARS-CoV-2 neutralizing FRNT antibody assays with these samples using an infectious clone of SARS-CoV-2. Only the SARS-CoV-2 spike primed mice generated neutralizing antibodies to SARS-CoV-2 (Fig 2E). Neutralizing titers significantly increased compared to baseline after the boost immunization of SARS-CoV-2 spike (180 vs. 31, $P = 0.006$) and further increased following the second HKU1 spike boost (520 vs. 180, $P = 0.050$). These results suggested that HKU1 spike boosted neutralizing antibodies to SARS-CoV-2, although ongoing affinity maturation to SARS-CoV-2 spike in the germinal centers may have also contributed to these findings. In contrast, the HKU1 spike/SARS-CoV-2 prime-boosted mice did not generate neutralizing antibodies to SARS-CoV-2 following any administration of antigen (Fig 2F). Thus, recent prior exposure to the endemic HKU1 coronavirus adjuvanted spike protein completely impeded the development of SARS-CoV-2 neutralizing antibodies in mice upon SARS-CoV-2 spike protein administration, despite the presence of SARS-CoV-2 binding antibodies. These data indicate that the SARS-CoV-2 spike boost immunization predominantly boosted cross-reactive antibodies to the original priming (HKU1) antigen at shared, non-neutralizing epitopes.

## Results in children

Because our mouse data suggested that prior exposure to endemic coronaviruses may blunt serologic responses to SARS-CoV-2 by the phenomenon of original antigenic sin, we next measured the baseline seroprevalence of IgG antibodies to both SARS-CoV-2 and HKU1 spike proteins in healthy children (n = 14) using pre-pandemic sera. The healthy, asymptomatic children were recruited from the community to participate in a phlebotomy study between 2016–2018, and they were comprised of 10 females (71%), median age 8 years (IQR 2.3–11.8 years), 8 (57%) Black, 4 (29%) White, 2 (14%) other race, and 14 (100%) non-Hispanic ethnicity. The geometric mean antibody titers from healthy children were then compared to children hospitalized at Children's Healthcare of Atlanta with acute COVID-19 (n = 14) or MIS-C (n = 14) (Table 1 and Fig 3) using one-way analysis of variance (ANOVA) of log-transformed titers.

As expected, we found significantly higher SARS-CoV-2 spike IgG titers in children with acute COVID-19 (7727 vs. 1019, P<0.001) and MIS-C (46989 vs. 1019, P<0.001) compared to healthy controls (Fig 3A). Moreover, children with MIS-C had significantly higher SARS-CoV-2 spike IgG titers compared to those with acute COVID-19 (Fig 3A, P<0.001), as we have previously shown [13]. All healthy controls had detectable antibodies to HKU1 spike with a wide range of titers (GMT 7551, 95% CI 3317 to 17190, Fig 3B). We found no difference

**Table 1. Clinical and demographic characteristics of patient cohort of children hospitalized with acute COVID-19 or MIS-C at Children's Healthcare of Atlanta.**

| | COVID-19 (n = 14) | n | MIS-C (n = 14) | n | P-value |
|---|---|---|---|---|---|
| **Age, years, mean (SD)** | 10.9 (7.6) | 14 | 9.1 (3.9) | 14 | 0.436 |
| **Gender, female, n (%)** | 8 (57%) | 14 | 7 (50%) | 14 | 0.379 |
| **Race, n (%)** | | | | | **0.006** |
| Black | 4 (29%) | 14 | 12 (86%) | 14 | |
| White | 9 (64%) | 14 | 1 (7%) | 14 | |
| Declined | 1 (7%) | 14 | | 14 | |
| **Ethnicity, n (%)** | | | | | 0.065 |
| Hispanic | 5 (36%) | 14 | 1 (7%) | 14 | |
| Non-Hispanic | 9 (64%) | 14 | 13 (93%) | 14 | |
| **Disease Severity, n (%)** | | | | | **0.002** |
| Mild/moderate | 10 (71%) | 14 | 2 (14%) | 14 | |
| Severe | 4 (29%) | 14 | 12 (86%) | 14 | |
| **Labs, mean (SD)** | | | | | |
| WBC max, x$10^3$cells/μL | 11.7 (5.8) | 13 | 11.9 (6.2) | 14 | 0.932 |
| ALC min, cells/μL | 2262.9 (1703.4) | 13 | 1055.1 (673.1) | 14 | **0.021** |
| Platelets min, x$10^3$cells/μL | 262.2 (106.0) | 13 | 153.8 (84.4) | 14 | **0.007** |
| ESR max, mm/hr | 34.9 (36.2) | 7 | 53.6 (22.8) | 10 | 0.209 |
| Sodium min, mmol/L | 137.3 (2.1) | 13 | 133.1 (4.6) | 14 | **0.006** |
| Creatinine max, mg/dL | 0.6 (0.2) | 13 | 1.1 (0.7) | 14 | **0.02** |
| ALT max, U/L | 148.5 (396.6) | 13 | 71.9 (48.5) | 14 | 0.479 |
| BNP max, pg/mL | 21.5 (16.1) | 5 | 1774.1 (1260.3) | 13 | **0.008** |
| Troponin max, ng/mL | 0.015 (0.0) | 4 | 1.2 (1.7) | 10 | 0.193 |
| Ferritin max, ng/mL | 217.5 (134.2) | 10 | 1385.4 (1278.5) | 14 | **0.009** |
| CRP max, mg/dL | 6.8 (7.5) | 12 | 19.5 (10.4) | 14 | **0.002** |
| **Imaging** | | | | | |
| CXR, n (%) | | | | | |
| Infiltrates | 3 (33%) | 9 | 10 (83%) | 12 | **0.02** |
| Pleural effusions | 3 (33%) | 9 | 9 (75%) | 12 | 0.056 |
| Echocardiogram, n (%) | | | | | |
| Depressed function | 0 (0%) | 4 | 7 (50%) | 14 | 0.07 |
| Coronary artery dilation | 0 (0%) | 4 | 2 (14%) | 14 | 0.423 |
| Treatment, n (%) | | | | | |
| Remdesivir | 4 (29%) | 14 | 1 (7%) | 14 | 0.139 |
| IVIG | 0 (0%) | 14 | 13 (93%) | 14 | <**0.001** |
| Steroids | 5 (36%) | 14 | 12 (86%) | 14 | **0.007** |
| Antiplatelet | 0 (0%) | 14 | 13 (93%) | 14 | <**0.001** |
| **Outcomes** | | | | | |
| Days of hospitalization, mean (SD) | 6.1 (7.8) | 14 | 9.4 (6) | 14 | 0.221 |
| ICU admission, n (%) | 5 (36%) | 14 | 12 (86%) | 14 | **0.007** |
| Days of ICU, mean (SD) | 2.6 (5.6) | 14 | 5.8 (5) | 14 | 0.123 |
| Low-flow $O_2$, n (%) | 5 (36%) | 14 | 11 (79%) | 14 | **0.022** |
| Mechanical ventilation, n (%) | 0 (0%) | 14 | 2 (14%) | 14 | 0.142 |
| Vasopressors, n (%) | 0 (0%) | 14 | 10 (71%) | 14 | <**0.001** |
| Death, n (%) | 0 (0%) | 14 | 0 (0%) | 14 | --- |

Max, maximum value obtained during the hospitalization; Min, minimum value obtained during the hospitalization; WBC, white blood cell count; ALC, absolute lymphocyte count; ESR, erythrocyte sedimentation rate; ALT, alanine aminotransferase; BNP, brain natriuretic peptide; CRP, C-reactive protein; CXR, chest radiograph; IVIG, intravenous immunoglobulin; ICU, intensive care unit; $O_2$, oxygen.

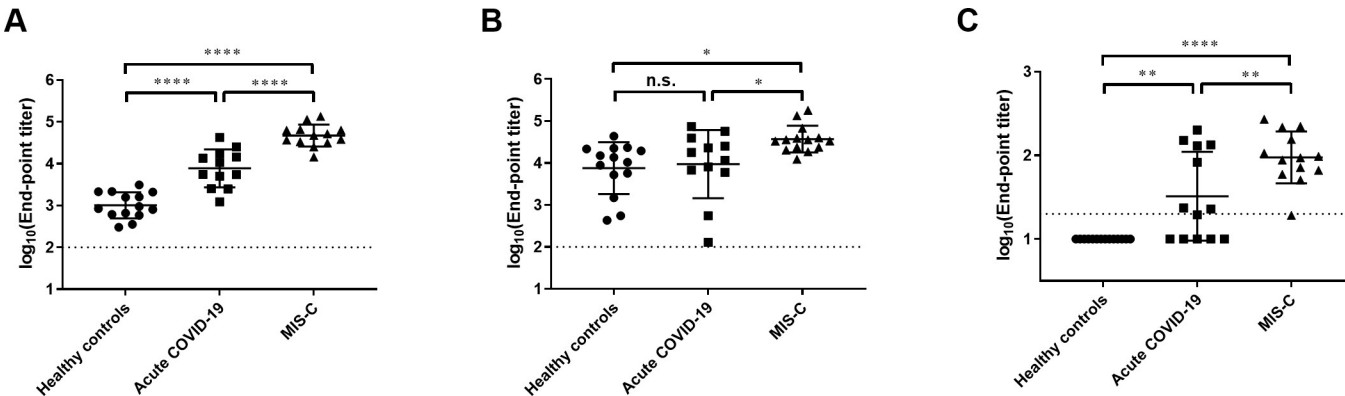

**Fig 3. HKU1 antibodies are prevalent in healthy children and children with acute COVID-19 and MIS-C.** SARS-CoV-2 (A) and HKU1 (B) spike IgG antibody titers and FRNT neutralization titers (C) in healthy pediatric controls compared to children hospitalized with acute COVID-19 and MIS-C. Each dot represents a unique patient, and data represent the geometric mean titers of *$P<0.05$; **$P<0.01$; ***$P<0.005$; ****$P<0.001$.

in HKU1 spike IgG titers between healthy controls and children with COVID-19 (7551 vs. 9376, $P = 0.917$), but children with MIS-C had significantly higher HKU1 spike IgG titers compared with healthy children (37239 vs. 7551, $P = 0.012$) and those with acute COVID-19 (37239 vs. 9376, $P = 0.043$, respectively).

We then tested the ability of plasma from the groups of children to neutralize an infectious clone of the SARS-CoV-2 virus using the FRNT assay. Plasma from healthy controls failed to neutralize SARS-CoV-2. In contrast, plasma from children with acute COVID-19 (33 vs. 10 (lower LOD), $P = 0.002$) and MIS-C (94 vs. 10, $P<0.001$) had significantly higher SARS-CoV-2 neutralization titers compared to healthy children (Fig 3C). The variability in neutralizing antibody titers observed in children with acute COVID-19 was likely attributable to the timing of sample collection post-symptom onset (POS) (median 6 days, IQR 4–11 days). In adults, the average time to detection of neutralizing antibodies against SARS-CoV-2 is 14.3 days POS with a wide range (range 3–59 days) [14], and these factors are likely reflected in our results. Children with MIS-C, which typically occurs 2–6 weeks post-COVID-19, had significantly higher neutralizing titers compared to children with acute COVID-19 ($P = 0.005$). In summary, although HKU1 spike antibodies were prevalent in healthy children with acute COVID-19 and MIS-C were able to mount strong neutralizing antibody responses to SARS-CoV-2.

We next performed multiple linear regression analyses and determined the Spearman correlations between the serologic assays among children with acute COVID-19 and MIS-C. We found that HKU1 spike IgG binding antibodies correlated strongly with SARS-CoV-2 spike binding antibody titers in children with acute COVID-19 and MIS-C ($r = 0.7577$, $P<0.001$) (Fig 4A). HKU1 spike IgG antibodies also correlated positively with SARS-CoV-2 neutralizing antibodies ($r = 0.6201$, $P = 0.001$) (Fig 4B), although the correlation was not as strong as that of SARS-CoV-2 spike IgG with neutralizing antibodies ($r = 0.7599$, $P<0.001$) (Fig 4C). Thus, our clinical data demonstrated that HKU1 spike antibodies are prevalent in healthy children, that they are higher in titer in children with MIS-C, and that they correlate with both SARS-CoV-2 spike binding and neutralizing antibodies in acute COVID-19 and MIS-C.

## Discussion

In this study, we found that priming mice with an endemic coronavirus HKU1 spike protein impeded the development of neutralizing antibodies to SARS-CoV-2 upon challenge with SARS-CoV-2 spike, consistent with original antigenic sin (OAS). Boosting mice

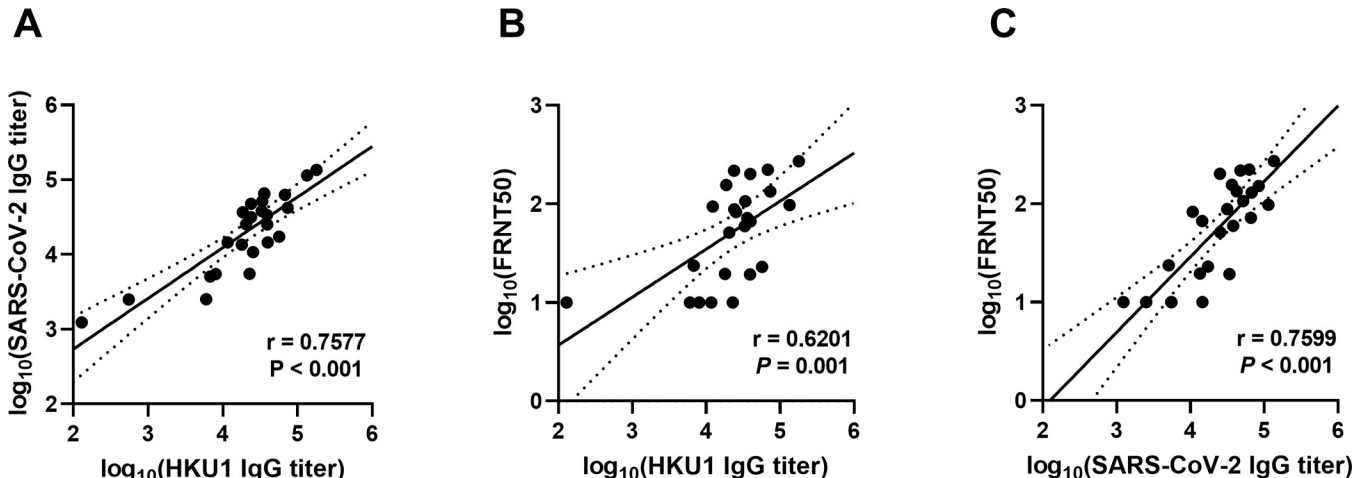

**Fig 4. HKU1 spike IgG antibodies correlated positively with both SAR-CoV-2 spike IgG and SARS-CoV-2 neutralizing antibodies in children with acute COVID-19 and MIS-C.** Linear regression analyses compared the log-transformed antibody titers of (A) SARS-CoV-2 spike IgG vs. HKU1 spike IgG; (B) HKU1 spike IgG vs. SARS-CoV-2 neutralization titers; and (C) SARS-CoV-2 spike IgG vs. SARS-CoV-2 neutralization titers among children with acute COVID-19 or MIS-C. Spearman's correlation coefficients (r) are shown.

intramuscularly with the heterologous coronavirus adjuvanted spike protein appeared to preferentially boost antibodies directed at the priming spike protein at shared, non-neutralizing epitopes. The phenomenon of OAS has been described in humans and animals with influenza [15, 16], dengue [17, 18], and human rhinovirus [19]. The pathophysiology of OAS is thought to be attributable to sequential exposure to similar, but non-identical antigens which preferentially elicits antibodies to the original antigen due to immunologic memory B cells [9]. In our study, long-lived plasma cells may have also contributed to the antibody responses observed following early boosting (3 weeks post-prime). Our data provides evidence that OAS plays a role in the murine serologic response to the spike protein of coronaviruses upon sequential exposure.

In contrast, analysis of clinical specimens from children with COVID-19 and MIS-C suggests that the serologic responses to SARS-CoV-2 in children may be amoral. Healthy pediatric controls had a wide range of pre-existing, binding antibody titers to HCoV HKU1 spike, reflecting a high baseline seroprevalence in children. Nevertheless, children with acute COVID-19 and MIS-C were able to effectively mount a neutralizing antibody response that correlated positively with both SARS-CoV-2 and HKU1 spike IgG antibodies. Thus, there was no direct evidence of OAS observed in our pediatric cohort. These results are consistent with previously published data, which indicate that SARS-CoV-2 infection elicits antibodies that cross-react with shared epitopes of endemic coronavirus spike proteins, predominantly within the conserved S2 subunit [20]. The discrepancy we observed with the experimental model of adjuvanted antigen administration in mice may be in part because the human immune system is capable of overcoming OAS, dependent upon factors relative to the original and subsequent antigen exposures (e.g. timing, duration, and magnitude).

To date, the literature describing the role of pre-existing HCoV immunity on SARS-CoV-2 serologic and clinical responses has been conflicting. Ng, et al. found that SARS-CoV-2 spike antibodies were prevalent in pre-pandemic sera, and that these were primarily IgG antibodies directed against the S2 subunit [8]. Interestingly, these cross-reactive antibodies possessed some neutralizing activity against SARS-CoV-2 pseudotyped viruses. This data conflicts with the pre-print findings of Anderson, et al. which demonstrated that endemic HCoV antibodies

are boosted by SARS-CoV-2 infection, but are not associated with neutralization or protection against SARS-CoV-2 infections or hospitalizations [21]. However, Sagar, et al. found that although recent endemic coronavirus infections did not impact susceptibility to SARS-CoV-2 infections or hospitalizations, they were associated with significant improvements in patient outcomes including mortality [22]. Thus, the role of pre-existing antibodies to endemic coronaviruses in SARS-CoV-2 immune responses and outcomes is incompletely understood.

Limitations of this study include the small number of mice in each group, and the limited number of available clinical samples. Importantly, we also lacked pre- and post-COVID-19 sera in a single patient cohort to definitively answer the question of the effects of pre-existing HCoV antibodies on SARS-CoV-2 acquisition and clinical outcomes. Population-based studies may provide greater insights into subtler effects of pre-existing HCoV cross-reactive immunity on SARS-CoV-2 infection. We only analyzed serologic immunity to one endemic coronavirus (HKU1 in the *Betacoronavirus* genus), and differences could exist among *Alphacoronaviruses* (229E and NL63) or with the other endemic *Betacoronavirus* (OC43). While HKU1 infection is less prevalent than NL63 and OC43, it shares more homology in the spike protein with SARS-CoV-2 compared to the other HCoVs [23, 24], so we chose to evaluate it for this reason in these experiments. Pre-existing immunity to other SARS-CoV-2 antigens such as the nucleocapsid protein may contribute to clinical response, but we did not evaluate these antibodies in this study. The converse question of whether SARS-CoV-2 antibodies could blunt serologic responses to endemic coronaviruses or to emerging SARS-CoV-2 variants remains to be determined.

## Conclusions

Prior exposure to endemic coronaviruses may blunt serologic responses to SARS-CoV-2 in mice by impeding the development of neutralizing antibodies, consistent with original antigenic sin. In comparison, our data in children suggest an amoral immune response. Future studies are needed to determine the effects of pre-existing immunity to endemic coronaviruses on clinical outcomes in children with acute COVID-19 and MIS-C.

## Experimental methods

### Mouse experiments

Six to 8-week-old female BALB/c mice were obtained from Jackson Laboratory (Bar Harbor, ME) and housed in pathogen-free conditions with 5 mice per cage. Animals were allowed to acclimate to the environment for 2 weeks prior to experimentation. All animal experiments were conducted according to approved protocols by the Emory University Institutional Animal Care and Use Committee (PROTO202000026). Treatment regimens described below were randomly assigned by cage, and investigators were not blinded to the group assignments or results.

One cage of Balb/c mice (n = 5) was primed and boosted at 21 days IM with 10 μg SARS-CoV-2 nucleocapsid protein (SinoBiological, 40588-V08B) in 50 μl with alum (Alhydrogel adjuvant 2%, Invivogen). The purpose of this was to generate polyclonal antiserum to the nucleocapsid protein early during the COVID-19 pandemic, which was not a part of the present study. These mice were immunized 3 weeks later analogously with 10 μg alum-adjuvanted HCoV-HKU1 S1+S2 ECD-His (SinoBiological, 40606-V08B) IM, followed by an identical boost 21 days later. Three weeks later the same mice were immunized with 10 μg alum-adjuvanted SARS-CoV-2 S1+S2 ECD-His (SinoBiological, 40589-V08B1) IM, followed by a final boost 21 days later. Plasma was collected by submandibular bleeding on day 21 following each administration. A terminal bleed via cardiac puncture was conducted 21 days after the final boost.

A second cage of mice (n = 5) was immunized with 10 μg alum-adjuvanted SARS-CoV-2 S1 +S2 ECD-His followed by a boost 21 days later. The mice were immunized 3 weeks later with 10 μg alum-adjuvanted HCoV-HKU1 S1+S2 ECD-His, followed by a boost 21 days later. Blood was collected as described above. Serum samples from each group were pooled and all serologic analyses were performed on pooled samples in duplicate. A small convenience sample size was chosen for this initial pilot study in mice. All animals (n = 10) were included in the final analyses, with no exclusions.

## Human subjects

Children and adolescents, 0 to 21 years of age, hospitalized at Children's Healthcare of Atlanta (CHOA) with confirmed or suspected COVID-19 or MIS-C were enrolled into an IRB-approved specimen collection protocol (Emory University IRB protocols STUDY00022371 and STUDY00000723) following written or verbal informed consent and assent as appropriate for age as previously described [13]. The Emory University IRB approved obtaining verbal consent and assent under specific circumstances, including to limit staff exposure to COVID-19. If verbal consent and assent were obtained, this was documented on the approved ICF form by the staff member who obtained consent. Children were classified as having MIS-C if they met the Centers for Disease Control and Prevention case definition [25]. They were classified as having acute COVID-19 if they were hospitalized with symptomatic disease and had SARS-CoV-2 detected by nasopharyngeal (NP) real-time polymerase chain reaction (RT-PCR). Prospective blood samples were collected from patients at the time of enrollment, and residual samples leftover from clinical labs were also obtained from the clinical laboratory. Plasma from healthy pediatric controls was collected through a separate IRB-approved protocol (STUDY0008846) following written informed consent and assent, as appropriate for age. Partial non-HKU1 serologic data from a subset of these children (4 with acute COVID-19 and 7 with MIS-C) were included in a prior publication [13]. Of the 126 planned analyses (3 groups, 14 patients per group, 3 types of serologic assays), there was insufficient volume to complete 5 assays. Data is available in the Supporting information.

## Serology

Recombinant SARS-CoV-2 S1+S2 ECD-His (SinoBiological, 40589-V08B1) and HCoV-HKU1 S1+S2 ECD-His (SinoBiological, 40606-V08B) were coated onto Nunc MaxiSorp plates at a concentration of 0.5 μg/mL in 100 μL phosphate-buffered saline (PBS) at 4°C overnight. Plates were blocked for two hours at room temperature in PBS/ 1% BSA/0.05% Tween-20 (ELISA buffer). Serum or plasma samples were heated to 56°C for 30 min, aliquoted, and stored at -20°C before use. Samples were serially diluted 1:3 in dilution buffer (PBS/1% BSA/0.05% Tween-20) starting at a dilution of 1:100. Coated plates were washed 4 times with 300 μl PBS/0.05% Tween-20 before adding 100 μL of each dilution and incubated for 90 minutes at room temperature. Plates were washed 4 times with PBS/0.05% Tween-20, and 100 μL of horseradish peroxidase-conjugated anti-Fc IgG antibody (Jackson ImmunoResearch Laboratories, 109-035-098), diluted 1:5,000 in ELISA buffer, was added and incubated for 60 minutes at room temperature. Plates were washed 4 times with PBS/0.05% Tween-20, followed by one additional wash with 1X PBS. Development was performed using 0.4 mg/mL o-phenylenediamine substrate (Sigma) in 0.05 M phosphate-citrate buffer pH 5.0, supplemented with 0.012% hydrogen peroxide before use. Reactions were incubated for 5 minutes then stopped with 1 M HCl and absorbance was measured at 490 nm.

## Focus reduction neutralization assays

FRNT assays were performed as previously described [11]. Briefly, samples were diluted at 3-fold in 8 serial dilutions using DMEM (VWR, #45000–304) in duplicates with an initial dilution of 1:10 in a total volume of 60 µl. Serially diluted samples were incubated with an equal volume of an infectious clone of SARS-CoV-2 (Wuhan strain, A.1 from PANGO lineage) (100–200 foci per well) at 37˚C for 1 hour in a round-bottomed 96-well culture plate. The antibody-virus mixture was then added to Vero cells and incubated at 37˚C for 1 hour. Post-incubation, the antibody-virus mixture was removed and 100 µl of prewarmed 0.85% methylcellulose (Sigma-Aldrich, #M0512-250G) overlay was added to each well. Plates were incubated at 37˚C for 24 hours. After 24 hours, methylcellulose overlay was removed, and cells were washed three times with PBS. Cells were then fixed with 2% paraformaldehyde in PBS (Electron Microscopy Sciences) for 30 minutes. Following fixation, plates were washed twice with PBS and 100 µl of permeabilization buffer (0.1% BSA [VWR, #0332], Saponin [Sigma, 47036-250G-F] in PBS), was added to the fixed Vero cells for 20 minutes. Cells were incubated with an anti-SARS-CoV spike primary antibody directly conjugated to biotin (CR3022-biotin) for 1 hour at room temperature. Next, the cells were washed three times in PBS and avidin-HRP was added for 1 hour at room temperature followed by three washes in PBS. Foci were visualized using TrueBlue HRP substrate (KPL, # 5510–0050) and imaged on an ELISPOT reader (CTL).

## Statistical analysis

Statistical comparisons were made with GraphPad Prism (v9.0). Antibody geometric mean titers (GMTs) of replicates were determined and log-transformed titers were compared using one-way analysis of variance (ANOVA) with Tukey's post-hoc comparison. Linear regression was performed on log-transformed antibody titers, and the Spearman's correlation coefficients (r) were calculated. $P$ values $\leq 0.05$ were considered statistically significant. Raw data with statistical comparisons and 95% confidence intervals are shown in the S1 Dataset.

## Supporting information

**S1 Dataset. Antibody titers in murine and human samples.**
(XLSX)

**S1 File. The ARRIVE essential 10: Compliance questionnaire.**
(PDF)

## Acknowledgments

We thank clinical research coordinators Beena Desai, Kerry Dibernardo, Felicia Glover, Vikash Patel, Maureen Richardson, Amber Samuel, and clinical research nurses Lisa Macoy and Kathy Stephens for their assistance enrolling patients and collecting specimens. We thank Nadine Rouphael and the Hope Clinic laboratory, Theda Gibson, Hui-Mien Hsiao, Wensheng Li, and the Emory Vaccine Research Center laboratory for their assistance processing specimens. We thank the Children's Healthcare of Atlanta research laboratory for their assistance in collecting residual specimens. We thank the professional staff of Emory Division of Animal Resources for their assistance with animal studies. Lastly, we thank the clinical study participants and their families for generously donating their time and samples to further our understanding of COVID-19 and MIS-C in children.

## Author Contributions

**Conceptualization:** Stacey A. Lapp, Larry J. Anderson, Evan J. Anderson, Christina A. Rostad.

**Data curation:** Stacey A. Lapp, Venkata Viswanadh Edara, Austin Lu, Laila Hussaini, Mehul S. Suthar, Evan J. Anderson, Christina A. Rostad.

**Formal analysis:** Stacey A. Lapp, Venkata Viswanadh Edara, Ann Chahroudi, Mehul S. Suthar, Evan J. Anderson, Christina A. Rostad.

**Funding acquisition:** Ann Chahroudi, Larry J. Anderson, Evan J. Anderson, Christina A. Rostad.

**Investigation:** Stacey A. Lapp, Venkata Viswanadh Edara, Lilin Lai, Mehul S. Suthar, Christina A. Rostad.

**Methodology:** Stacey A. Lapp, Mehul S. Suthar, Evan J. Anderson, Christina A. Rostad.

**Project administration:** Laila Hussaini.

**Resources:** Ann Chahroudi, Evan J. Anderson, Christina A. Rostad.

**Supervision:** Laila Hussaini, Mehul S. Suthar, Evan J. Anderson.

**Writing – original draft:** Stacey A. Lapp, Christina A. Rostad.

**Writing – review & editing:** Stacey A. Lapp, Venkata Viswanadh Edara, Austin Lu, Lilin Lai, Laila Hussaini, Ann Chahroudi, Larry J. Anderson, Mehul S. Suthar, Evan J. Anderson, Christina A. Rostad.

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
