## [Decision Letter · Decision Letter 0]

12 May 2021

PONE-D-21-10490

Original antigenic sin responses to heterologous Betacoronavirus spike proteins are observed in mice following intramuscular administration, but are not apparent in children following SARS-CoV-2 infection

PLOS ONE

Dear Dr. Rostad,

Thank you for submitting your manuscript to PLOS ONE. After careful consideration, we feel that it has merit but does not fully meet PLOS ONE’s publication criteria as it currently stands. Therefore, we invite you to submit a revised version of the manuscript that addresses the points raised during the review process.

During the review process, substantial concerns were brought up regarding experimental design and presentation of results that ultimately reduce confidence that the data presented allow for the conclusions drawn.  It may be difficult to address these concerns during revision, as models may have to be re-designed and the conclusions revisited based on the interpretation of any new data collected.  During the revision process, please specifically address the comments related to experimental design, interpretation of results, and the conclusions drawn as these will be critical for determining the validity of the findings in the context of OAS and SARS-CoV-2.

We look forward to receiving your revised manuscript.

Kind regards,

Victor C Huber

Academic Editor

PLOS ONE

Journal Requirements:

2. As part of your revision, please complete and submit a copy of the ARRIVE Guidelines checklist, a document that aims to improve experimental reporting and reproducibility of animal studies for purposes of post-publication data analysis and reproducibility: https://www.nc3rs.org.uk/arrive-guidelines. Please include your completed checklist as a Supporting Information file. Note that if your paper is accepted for publication, this checklist will be published as part of your article.

[I have read the journal's policy and the authors of this manuscript have the following competing interests:

E.J.A. has received personal fees from AbbVie, Pfizer, and Sanofi Pasteur for consulting, and his institution receives funds to conduct clinical research unrelated to this manuscript from MedImmune, Regeneron, PaxVax, Pfizer, GSK, Merck, Novavax, Sanofi-Pasteur, Janssen, and Micron. He also serves on a safety monitoring board for Sanofi-Pasteur and Kentucky BioProcessing, Inc.

C.A.R.’s institution has received funds to conduct clinical research unrelated to this manuscript from BioFire Inc, GSK, MedImmune, Micron, Janssen, Merck, Moderna, Novavax, PaxVax, Pfizer, Regeneron, Sanofi-Pasteur. She is co-inventor of patented RSV vaccine technology unrelated to this manuscript, which has been licensed to Meissa Vaccines, Inc.].

4. We noted in your submission details that a portion of your manuscript may have been presented or published elsewhere.

[Partial non-HKU1 serologic data from a subset of these children (4 with acute COVID-19 and 7 with MIS-C) were included in a prior publication: https://pediatrics.aappublications.org/content/pediatrics/146/6/e2020018242.full.pdf]

Reviewers' comments:

Reviewer's Responses to Questions

**Comments to the Author**

1. Is the manuscript technically sound, and do the data support the conclusions?

Reviewer #1: No

Reviewer #2: Partly

2. Has the statistical analysis been performed appropriately and rigorously? 

Reviewer #1: Yes

Reviewer #2: Yes

3. Have the authors made all data underlying the findings in their manuscript fully available?

Reviewer #1: Yes

Reviewer #2: Yes

4. Is the manuscript presented in an intelligible fashion and written in standard English?

Reviewer #1: Yes

Reviewer #2: Yes

5. Review Comments to the Author

Reviewer #1: Lapp and colleagues consider the important question of whether pre-existing immunity to endemic human coronaviruses impacts the Ab response to SARS-CoV-2 infection. Specifically, they investigate whether previous B cell responses to the spike (S) protein of the endemic betacoronavirus HKU1 modulate Ab production against the antigenically-related S protein of SARS-CoV-2. The authors postulate that the response might reflect original antigenic sin (OAS) and feature preferential induction of Abs against the original priming antigen, the HKU1 S protein, diminishing Ab production against the SARS-CoV-2 S protein. The study is in two parts: (i) a mouse model study using IM immunization with the HKU1 and SARS-CoV-2 S proteins to test for OAS Ab responses, and (ii) analysis of serum Abs to HKU1 and SARS-CoV-2 S proteins in pre-pandemic children and hospitalized children with acute COVID-19 or MIS-C. The authors conclude from the mouse study that OAS impedes the Ab response to SARS-CoV-2 S after priming with HKU1 S, but that OAS does not appear to be a factor in the Ab responses of children to SARS-CoV-2 infection.

The study is clearly described and well justified. However, I have concerns about conclusions drawn from the data, as described under the following points.

1. In the mouse studies, IgG levels generated in response to SARS-CoV-2 S did not further increase after the HKU1 S boost (Fig. 2A). The result was the same in the reciprocal regimen: priming with HKU1 S and boosting with SARS-CoV-2 S (Fig. 2D). In these experiments, the boost did not elicit a response primarily against the priming antigen. Why was there no OAS response in these experiments, since other experiments (Figs 2B and C) demonstrate cross-reactivity between the S of HKU1 and SARS-CoV-2? In fact, Figs 2B and C indicate that the boost increased titers to the boosting antigen rather than to the priming antigen. Could the protein administered in the boost drive affinity maturation against related epitopes on the boosting antigen?

2. Fig. 2E: The authors conclude that HKU1 S boosted neutralizing Abs to SARS-CoV-2. Why is this “OAS” boost not evident in Fig. 2A? Could the result in Fig. 2E reflect an ongoing germinal center response (ongoing affinity maturation) to the priming SARS-CoV-2 S? Note that 10 micrograms given IM with alum is a relatively large dose with a depot adjuvant that could potentially maintain germinal center reactions for weeks/months. Could the same results be obtained without the HKU1 boosts?

3. Fig. 2F: The authors state that the HKU1 S immunization completely impeded development of SARS-CoV-2 neutralizing Abs on SARS-CoV-2 S immunization (line 138) and suggest that the SARS-CoV-2 S predominantly boosted cross-reactive Abs to the priming HKU1 S at shared non-neutralizing epitopes (line 140). This is hard to reconcile with the suggestion that HKU1 S boosted neutralizing Abs to SARS-CoV-2 (line 135), presumably by boosting Abs to shared neutralizing epitopes. Furthermore, why is boosting of any Abs to the priming HKU1 S by SARS-CoV-2 S not evident in Fig. 2D?

4. The analysis performed on sera from healthy children (pre-pandemic) and children with acute COVID-19 and MIS-C does not provide any insight into OAS effects on the Ab response to SARS-CoV-2 S. Multiple studies have demonstrated that SARS-CoV-2 infection generates Abs to the conserved S2 of the SARS-CoV-2 S that cross-react with seasonal betacoronavirus S proteins, as well as neutralizing Abs to novel S1 epitopes. Hence, the described correlations (Fig. 4) are expected and do not exclude an anti-S response that has OAS features and occurs early in the response.

Reviewer #2: The authors present a study to investigate the effects of original antigenic sin on the development of the adaptive immune response to SARS-CoV-2. Overall, the methodology and analysis are satisfactory. I have concerns regarding the interpretation of the results and the extent to which the data support the conclusions drawn.

1) The authors point out a discrepancy in their results in the mouse model and the results from analyzing sera from children. Specifically, results were consistent with OAS in the mouse model but not in the serological study. In the mouse study, the time between the last HKU exposure and first SARS-CoV-2 exposure was approximately 3 weeks. In general, imprinting effects are attributed to the recall of memory B-cells, but this short amount of time between exposures could mean that long-lived plasma cells might be playing an important role in clearing antigen before memory recall could occur. Can the authors address the extent to which the observed effects might be due to long-lived plasma cells as opposed to memory?

2) For the serological study, the authors show that children are able to mount a neutralizing antibody response to SARS-CoV-2 despite having high titers to HKU. However, looking at Figure 3C, although the group with acute infection does have higher mean neutralization titer than the control group, it appears as though most of the points have nearly undetectable or undetectable titer. Was the time between symptom onset and subject enrollment relatively consistent across the acutely-infected group, and could that explain the observed variance in neutralizing titer? Is it possible that not all acutely infected individuals had a previous HKU infection? In general, I find it hard to evaluate how the serological results are/are not consistent with OAS given the uncertainty of prior HKU exposure and the lack of specific information on the timing of patient sampling.

6. PLOS authors have the option to publish the peer review history of their article (what does this mean?). If published, this will include your full peer review and any attached files.

Reviewer #1: No

Reviewer #2: No

---

## [Author Response · Author response to Decision Letter 0]

8 Jul 2021

We thank the reviewers for their time and feedback to provide us with the opportunity to strengthen our manuscript. Please find below our point-by-point responses.

Review Comments to the Author

Reviewer #1: Lapp and colleagues consider the important question of whether pre-existing immunity to endemic human coronaviruses impacts the Ab response to SARS-CoV-2 infection. Specifically, they investigate whether previous B cell responses to the spike (S) protein of the endemic betacoronavirus HKU1 modulate Ab production against the antigenically-related S protein of SARS-CoV-2. The authors postulate that the response might reflect original antigenic sin (OAS) and feature preferential induction of Abs against the original priming antigen, the HKU1 S protein, diminishing Ab production against the SARS-CoV-2 S protein. The study is in two parts: (i) a mouse model study using IM immunization with the HKU1 and SARS-CoV-2 S proteins to test for OAS Ab responses, and (ii) analysis of serum Abs to HKU1 and SARS-CoV-2 S proteins in pre-pandemic children and hospitalized children with acute COVID-19 or MIS-C. The authors conclude from the mouse study that OAS impedes the Ab response to SARS-CoV-2 S after priming with HKU1 S, but that OAS does not appear to be a factor in the Ab responses of children to SARS-CoV-2 infection. The study is clearly described and well justified. However, I have concerns about conclusions drawn from the data, as described under the following points.

1. In the mouse studies, IgG levels generated in response to SARS-CoV-2 S did not further increase after the HKU1 S boost (Fig. 2A). The result was the same in the reciprocal regimen: priming with HKU1 S and boosting with SARS-CoV-2 S (Fig. 2D). In these experiments, the boost did not elicit a response primarily against the priming antigen. Why was there no OAS response in these experiments, since other experiments (Figs 2B and C) demonstrate cross-reactivity between the S of HKU1 and SARS-CoV-2? In fact, Figs 2B and C indicate that the boost increased titers to the boosting antigen rather than to the priming antigen. Could the protein administered in the boost drive affinity maturation against related epitopes on the boosting antigen?

The authors thank the reviewer for this interesting question. Antibody and B cell responses following repeated vaccination/administration peak and plateau, such that further boosting does not further increase antibody titers. The mechanism for this has been attributed to “antibody feedback,” such that high titers of antibodies mask the immunodominant epitopes and limit further B cell expansion. Repeated boosting is thought to diversify the antibody responses, as subdominant responses expand. This was recently described for human immune responses to malaria vaccine (McNamara HA, et al. Cell. July 2020). We have added the text in the results section to describe this (Line 130): “The plateau in antibody titers observed following repeated boosting is a known phenomenon whereby antibody feedback limits further B cell expansion by masking immunodominant epitopes.”

Our findings do suggest that heterologous boosting drives affinity maturation against related epitopes on the boosting antigen, as seen in Figures 2B and 2C. But the fact that these antibodies lack neutralization activity indicates that they are predominantly cross-reactive antibodies directed against the original priming strain at shared, but non-neutralizing epitopes. We have modified the text to clarify this point on Line 120: “These findings suggested that heterologous boosting drove affinity maturation against related epitopes on the boosting SARS-CoV-2 spike antigen.” And Line 142: “… recent prior exposure to the endemic HKU1 coronavirus adjuvanted spike protein completely impeded the development of SARS-CoV-2 neutralizing antibodies in mice upon SARS-CoV-2 spike protein administration, despite the presence of SARS-CoV-2 binding antibodies. These data indicate that the SARS-CoV-2 spike boost immunization predominantly boosted cross-reactive antibodies to the original priming (HKU1) antigen at shared, non-neutralizing epitopes.”

2. Fig. 2E: The authors conclude that HKU1 S boosted neutralizing Abs to SARS-CoV-2. Why is this “OAS” boost not evident in Fig. 2A? 

We thank the reviewer for this question. The magnitude of binding antibodies have peaked and plateaued following repeated boosting in Fig. 2A, as described above. Further boosting allows for diversification of antibody response as non-dominant epitopes are exposed. 

Could the result in Fig. 2E reflect an ongoing germinal center response (ongoing affinity maturation) to the priming SARS-CoV-2 S? Note that 10 micrograms given IM with alum is a relatively large dose with a depot adjuvant that could potentially maintain germinal center reactions for weeks/months. Could the same results be obtained without the HKU1 boosts?

This is an interesting point, and we cannot exclude that ongoing affinity maturation to SARS-CoV-2 spike in the germinal center is contributing to the increases in IgG titers following HKU1 boost. We have therefore modified the text accordingly on Line 135: “Neutralizing titers significantly increased compared to baseline after the boost immunization of SARS-CoV-2 spike (180 vs. 31, P=0.006) and further increased following the second HKU1 spike boost (520 vs. 180, P=0.050). These results suggested that HKU1 spike boosted neutralizing antibodies to SARS-CoV-2, although ongoing affinity maturation to SARS-CoV-2 spike in the germinal centers may have also contributed to these findings.”

3. Fig. 2F: The authors state that the HKU1 S immunization completely impeded development of SARS-CoV-2 neutralizing Abs on SARS-CoV-2 S immunization (line 138) and suggest that the SARS-CoV-2 S predominantly boosted cross-reactive Abs to the priming HKU1 S at shared non-neutralizing epitopes (line 140). This is hard to reconcile with the suggestion that HKU1 S boosted neutralizing Abs to SARS-CoV-2 (line 135), presumably by boosting Abs to shared neutralizing epitopes. 

Yes, to state these findings another way:

- HKU1 S prime immunization completely impeded the development of neutralizing antibodies to SARS-CoV-2 upon SARS-CoV-2 S boost immunization because the boosted antibodies were directed toward the priming (HKU1) strain at non-neutralizing epitopes for SARS-CoV-2. 

- By this same rationale, HKU1 S boost administered after SARS-CoV-2 prime elicited antibodies directed toward the priming (SARS-CoV-2) strain at neutralizing epitopes for SARS-CoV-2.

- In both cases, the boost elicited antibodies directed against cross-reactive epitopes on the priming antigen. The antibodies were only able to neutralize SARS-CoV-2 if the prime was SARS-CoV-2 spike.

Furthermore, why is boosting of any Abs to the priming HKU1 S by SARS-CoV-2 S not evident in Fig. 2D?

In figure 2D, the antibodies have peaked and plateaued, as described above (McNamara HA, et al. Cell. July 2020) due to antibody feedback. Further boosting is not expected to further increase antibody titers.

4. The analysis performed on sera from healthy children (pre-pandemic) and children with acute COVID-19 and MIS-C does not provide any insight into OAS effects on the Ab response to SARS-CoV-2 S. Multiple studies have demonstrated that SARS-CoV-2 infection generates Abs to the conserved S2 of the SARS-CoV-2 S that cross-react with seasonal betacoronavirus S proteins, as well as neutralizing Abs to novel S1 epitopes. Hence, the described correlations (Fig. 4) are expected and do not exclude an anti-S response that has OAS features and occurs early in the response.

We acknowledge that the human data does not directly prove or disprove the presence of OAS in children with COVID-19 or MIS-C, and we have discussed this in the limitations paragraph. We do think that the human data provides context for interpreting the animal studies, as we think it is helpful to show that HKU1 antibodies are prevalent in healthy children. Further, if OAS plays a role in human serologic responses, children with COVID-19 or MIS-C were nevertheless able to mount SARS-CoV-2 neutralizing antibodies, which correlated strongly with HKU1 spike IgG. We have modified the text to address the reviewer’s comments on Line 284: “These results are consistent with previously published data, which indicate that SARS-CoV-2 infection elicits antibodies that cross-react with shared epitopes of endemic coronavirus spike proteins, predominantly within the conserved S2 subunit (Ladner, Cell Reports Medicine, Jan 2021).”

Reviewer #2: The authors present a study to investigate the effects of original antigenic sin on the development of the adaptive immune response to SARS-CoV-2. Overall, the methodology and analysis are satisfactory. I have concerns regarding the interpretation of the results and the extent to which the data support the conclusions drawn.

1) The authors point out a discrepancy in their results in the mouse model and the results from analyzing sera from children. Specifically, results were consistent with OAS in the mouse model but not in the serological study. In the mouse study, the time between the last HKU exposure and first SARS-CoV-2 exposure was approximately 3 weeks. In general, imprinting effects are attributed to the recall of memory B-cells, but this short amount of time between exposures could mean that long-lived plasma cells might be playing an important role in clearing antigen before memory recall could occur. Can the authors address the extent to which the observed effects might be due to long-lived plasma cells as opposed to memory?

We thank the reviewer for this interesting question. We did not determine the extent to which long-lived plasma cells vs. memory B cells were contributing to OAS responses observed in the mice. Long-lived plasma cells can survive weeks to months following a murine exposure, so they likely did contribute to the responses we observed, and perhaps more so because of the early boost at 3 weeks post-prime. We have modified the discussion to take this into consideration in Line 274: “In our study, long-lived plasma cells may have also contributed to the antibody responses observed following early boosting regimen (3 weeks post-prime).”

2) For the serological study, the authors show that children are able to mount a neutralizing antibody response to SARS-CoV-2 despite having high titers to HKU. However, looking at Figure 3C, although the group with acute infection does have higher mean neutralization titer than the control group, it appears as though most of the points have nearly undetectable or undetectable titer. Was the time between symptom onset and subject enrollment relatively consistent across the acutely-infected group, and could that explain the observed variance in neutralizing titer? Is it possible that not all acutely infected individuals had a previous HKU infection? In general, I find it hard to evaluate how the serological results are/are not consistent with OAS given the uncertainty of prior HKU exposure and the lack of specific information on the timing of patient sampling.

The timing of sample collection post-onset of symptoms (POS) in the patients with COVID-19 was median 6 days (IQR 4-11 days). At this early time point POS, it is not surprising that some patients had not yet developed neutralizing antibodies. In adults, the average time to detection of neutralizing antibodies against SARS-CoV-2 is 14.3 days POS with a wide range (range 3-59 days) (Seow, Nature Microbiology, October 2020). In our cohort, it is likely that the variability in timing POS of sample collection, in addition to the intrinsic variability in timing of seroconversion, contributed to the observed variability in neutralizing titer. It is possible that not all acutely infected individuals had a previous HKU1 infection, although all had detectable HKU1 IgG antibodies. Nevertheless, we concur that our lack of pre-infection samples to ascertain prior HKU1 exposure is a limitation of the study.

We have included text in the manuscript to communicate this information on Line 230: “The variability in neutralizing antibody titers observed in children with acute COVID-19 was likely attributable to the timing of sample collection post-symptom onset (POS) (median 6 days, IQR 4-11 days). In adults, the average time to detection of neutralizing antibodies against SARS-CoV-2 is 14.3 days POS with a wide range (range 3-59 days) (Seow, Nature Microbiology, Oct 2020), and these factors are likely reflected in our results. Children with MIS-C, which typically occurs 2-6 weeks post-COVID-19, had significantly higher neutralizing titers compared to children with acute COVID-19 (P=0.005).” and Line 304: “Importantly, we also lacked pre- and post-COVID-19 sera in a single patient cohort to definitively answer the question of the effects of pre-existing HCoV antibodies on SARS-CoV-2 acquisition and clinical outcomes.”

---

## [Decision Letter · Decision Letter 1]

9 Aug 2021

Original antigenic sin responses to Betacoronavirus spike proteins are observed in a mouse model, but are not apparent in children following SARS-CoV-2 infection

PONE-D-21-10490R1

Dear Dr. Rostad,

We’re pleased to inform you that your manuscript has been judged scientifically suitable for publication and will be formally accepted for publication once it meets all outstanding technical requirements.

Kind regards,

Victor C Huber

Academic Editor

PLOS ONE

Additional Editor Comments (optional):

Reviewers' comments:

Reviewer's Responses to Questions

**Comments to the Author**

1. If the authors have adequately addressed your comments raised in a previous round of review and you feel that this manuscript is now acceptable for publication, you may indicate that here to bypass the “Comments to the Author” section, enter your conflict of interest statement in the “Confidential to Editor” section, and submit your "Accept" recommendation.

Reviewer #1: All comments have been addressed

Reviewer #2: All comments have been addressed

2. Is the manuscript technically sound, and do the data support the conclusions?

Reviewer #1: Yes

Reviewer #2: Yes

3. Has the statistical analysis been performed appropriately and rigorously? 

Reviewer #1: Yes

Reviewer #2: Yes

4. Have the authors made all data underlying the findings in their manuscript fully available?

Reviewer #1: Yes

Reviewer #2: Yes

5. Is the manuscript presented in an intelligible fashion and written in standard English?

Reviewer #1: Yes

Reviewer #2: Yes

6. Review Comments to the Author

Reviewer #1: I thank the authors for their thoughtful consideration of reviewer questions and their comprehensive responses. Revisions to the manuscript have clarified interpretation of findings and will be helpful to readers.

Reviewer #2: I am satisfied with the response to my previous comments and thank the authors for their thoughtfulness.

7. PLOS authors have the option to publish the peer review history of their article (what does this mean?). If published, this will include your full peer review and any attached files.

Reviewer #1: No

Reviewer #2: No

---

## [Editor Report · Acceptance letter]

20 Aug 2021

PONE-D-21-10490R1 

Original antigenic sin responses to *Betacoronavirus* spike proteins are observed in a mouse model, but are not apparent in children following SARS-CoV-2 infection 

Dear Dr. Rostad:

I'm pleased to inform you that your manuscript has been deemed suitable for publication in PLOS ONE. Congratulations! Your manuscript is now with our production department. 

Kind regards, 

on behalf of

Dr. Victor C Huber 

Academic Editor

PLOS ONE